# The interplay between eWOM information and purchase intention on social media: Through the lens of IAM and TAM theory

Md. Atikur Rahaman[1], H. M. Kamrul Hassan[2], Ahmed Al Asheq[3,4]*, K. M. Anwarul Islam[5]

1 School of Management, Jiujiang University, Jiangxi, China, 2 Department of Marketing, Faculty of Business Administration, University of Chittagong, Chattogram, Bangladesh, 3 School of Business, Maynooth University, Maynooth, Ireland, 4 Department of Business Administration, World University of Bangladesh (WUB), Dhaka, Bangladesh, 5 Department of Business Administration, The Millennium University, Dhaka, Bangladesh

* ahmed.asheq.2022@mumail.ie, aasheq@business.wub.edu.bd

**Data Availability Statement:** All relevant data are within the paper and its Supporting information files.

## Abstract

The maturity and growth of social media have empowered online customers to generate electronic word of mouth (eWOM), on various online websites and platforms, which may influence an individual's decision-making process. This paper explores eWOM information's impact on social media users' purchase intention by applying the information adoption model (IAM) and the technology acceptance model (TAM). PLS-SEM (SmartPLS V.3.3) has been utilized to test the hypotheses using data of 432 respondents. The research findings evinced that eWOM information quality, credibility, usefulness, and ease of use have been critical in determining online consumers' intention to adopt eWOM and form purchase behavior on social media. The study's outcomes offer the marketing managers a viewpoint to realize the significance of the effect of eWOM information on online purchase intention among social media users. Furthermore, the study findings will also enlighten marketing and business managers to utilize social media websites by gauging consumer behavior and focusing on characteristics of eWOM information on social media for better consumer insights.

## 1. Introduction

The swift development of technological progress has offered customers several modes of communication channels to interact with business firms in today's time [1]. Along with this technological advancement, various social media platforms have been developed to facilitate the exchange of information among online customers. Globally, social media is considered one of the most powerful platforms among consumers to share and spread information [2]. The maturity and growth of social media have also empowered online customers to generate electronic word of mouth, known as "eWOM," on various online websites and platforms, utilizing product/service reviews, blogs, recommendations, and so on [3]. Researchers [4] defined the

**Funding:** This study was supported by the School of Management, Jiujiang University and Center for Entreprenureship & Innovation Research Society (grant number: 43120227). The funders have no role in study design, data collection and analysis, decision to publish, or preparation of the manuscript. The funders will bear the cost of the article processing charge of the paper subject to the acceptance of the manuscript.

**Competing interests:** No conflict of interest among the authors.

concept of "eWOM" as "any positive or negative statement made by potential, actual or former customer which is available to a multitude of people via the internet" (p. 39). The eWOM information is more likely to be reliable as individuals can experience a neutral opinion about a product or service [5]. Therefore, eWOM information has been discerned as a more powerful medium to influence an individual's decision-making process [6]. Before purchasing a product or service, consumers tend to look for more information about the product online and significantly consider the review comments about the product generated by other consumers [7]. Consumers rely on eWOM information to minimize their doubts and risk when making a purchase effort [8]. Past research exhibited that eWOM impacted the consumers' selection process of goods/services [9] and influenced individuals' purchasing decisions [10].

Nonetheless, social media websites are considered a relatively new platform for generating eWOM, allowing online users to develop a connection with other users by sharing their opinions, ideas, and experiences [11]. Through social media engagement, people can now exchange their viewpoints and share experiences about a product/service with their friends and familiar people in the network [12]. This reciprocal information-sharing process in social media makes eWOM information more authentic and dependable for the consumers. The social media-generated conversations were found to have a robust natural impact on online purchase behavior [13]. It is observed that online individuals form their purchase decisions based on eWOM information. Henceforth, it is also critical for them to sort out and screen out the relevant information available online before using as online users are generally exposed to a boundless quantity of eWOM information on social media websites. Still, the aspects of eWOM information and online purchase intention at social media websites are not well explored in the literature. More specifically, the roles of eWOM information in triggering consumer behaviors such as buying intention have not been largely investigated in regard to the eWOM perspective [14–16]. Furthermore, others [17] argued that the influence of eWOM not only relies on the information but also depends on online users on social media. Hence, it is of importance to evaluate the influence of various aspects of eWOM information on consumer purchase intention from the social media perspective. To achieve this research aim, both Information Adoption Model (IAM) and Technology Acceptance Model (TAM) have been considered as it is believed that IAM is an appropriate model to illustrate the aspects of eWOM information. In contrast, TAM will explain online consumer behavior towards eWOM.

The current paper presents a number of valuable and noteworthy contributions to the current literature based on the consumer behavior perspective. Firstly, several studies have examined the impact of eWOM on consumer purchase intention by considering the TAM model [18, 19] in which factor: "perceived usefulness" of the TAM model was adopted as a determinant of information adoption and purchase intention. Still, the impact of perceived ease of use of eWOM on consumer purchase decisions was merely reported in the literature. The novelty of this study lies in attempting to fill out this gap by analyzing the influence of ease of use of eWOM on information adoption behavior and purchase intention. Secondly, this study has considered a recent growing economy: Bangladesh, to draw the study sample as it is observed that the country has been experiencing a digital transformation across the nation [20], where eWOM plays a critical role in developing the business growth, especially on online platforms. Lastly, several studies in Bangladesh were carried out to exhibit the connection of social media with brand equity [21], consumer behavior [22], and purchase intention [23, 24]. In contrast, the relationship between various attributes of eWOM and consumer behavioral aspects was rarely tested in the Bangladesh context, where social media-based businesses have started flourishing day by day. Henceforth, the current study has filled this void by integrating the IAM and TAM models to predict consumer purchase intention in Bangladesh. The findings of this paper also offer theoretical knowledge on the roles of eWOM information on consumer

behavior in social media and strive to contribute to the body of literature through the research model. Finally, as a means of managerial implications, it has been crucial to examine the impact of eWOM information on purchase intention in the social media context. Because it would offer online consumer insights to the marketing managers to develop a better marketing strategy by leveraging eWOM information.

## 2. Electronic word of mouth (eWOM) in social media

The eWOM information can be referred as any form of comment regarding a product or service, which is easily accessible and available to numerous individuals on internet-mediated platforms [25]. Social media websites have offered a new platform for the communication channel to disseminate information, and these websites are regarded as one of the suitable and appropriate platforms for electronic word of mouth (eWOM) [26]. On these websites, individual users can easily make any comments and share any information in the form of text writing, posting a photo or uploading a video, etc. Apparently, enriched contents on social media websites result in making eWOM information more noticeable and entertaining to online users [27]. On social media websites, online users usually like to read and check various comments, thoughts, and others' experiential opinions about products or services [2]. Therefore, online consumers are more intentional to resort to diverse platforms of social media in order to gather the needful information about brands [28].

Generally, eWOM information may emerge in several means on social media platforms, such as online users can deliberately share their opinions about brands by posting on social media, or they can reveal their liking for any brand by becoming a member of an online fan club. Even marketing people can purposefully share various helpful information about the brands on the official page of their social media accounts [29]. So, eWOM information might have an important influence on online consumers to decide which products they will purchase. As eWOM helps individuals to make better purchase decision [30], hence it is very critical to find out what aspects or characteristics of eWOM information might affect consumers' purchase intention on social media websites.

## 3. Literature review & hypotheses development

### 3.1. Theoretical underpinning

Several research studies have adopted the information adoption method to evaluate how individuals make use of the information they receive or the message they convey to themselves [31]. Many researchers in the discipline of information systems have adopted the concept of the "Information Adoption Model (IAM)" to demonstrate how humans receive persuasive information in order to make decisions [32, 33]. Researchers [34] proposed the theory of IAM, which was originally formulated on the basis of the integration of both the "Elaboration Likelihood Model (ELM)" [35] and the "Technology Acceptance Model (TAM)" [36]. This IAM model demonstrates that individuals are significantly influenced by a set of information in two different ways: either through the central path or through the peripheral path [37]. The core of the communication is shown by the central path, whereas the peripheral path is not directly related to the central part of the message [38]. The IAM is often combined of four attributes, which are argument quality (which corresponds to the central route), source credibility (which corresponds to the peripheral pathway), functionality and incorporation of the information (which refers to the periphery path), and overall effectiveness [39]. IAM aims to demonstrate how people tend to be impacted by computer-mediated information on internet-based platforms by addressing these four factors [10]. Thus, the IAM model notably determines the trustworthiness and quality of eWOM information.

Alternatively, the "Technology Acceptance Model (TAM)" developed by other scholars [36], is one of the most commonly used theories that truly describes any behavioural attributes of information technology (IT) users when they encounter with the task of embracing and acknowledging a modern IT-oriented platform [40, 41]. Despite the fact that this model was originally formed from the "Theory of Reasoned Action (TRA)," TAM theory has been continued to be based on information technology [42], in contrast to the way that TRA theory has tended to be restricted to behavioral models [43]. It is generally built on two constructs: perceived usefulness and perceived ease of use, both of which are utilized to forecast an individual's likelihood to embrace a technology system [44]. There has been a substantial application of this approach in various settings, including social media usage [45], mobile banking usage [46], and online review [47]. This study examines the usefulness and ease of use of eWOM information obtained via TAM to determine whether it impacts online purchase intention.

### 3.2. Hypotheses development

**3.2.1. The relationship of eWOM information quality (IQ) with perceived usefulness (USE) and ease of Use of eWOM Information (EOU).** Information quality (IQ) indicates the users' perception-based assessment of whether the given information attributes fulfill their usage purpose or meet up the needs of any given system [7]. Information quality (IQ) represents the correctness, lucidity, understandability, and dependability of the given information embedded into a system [48]. The scholars [49] described that consistency and completeness had been considered the two critical determining factors of information quality. According to researchers [50], "The issues of information quality and credibility are gaining importance, particularly in the World Wide Web context. The WWW provides unique information-seeking environment but often lacks quality control mechanisms" (p. 1243). IQ plays a crucial part in assessing and appraising the quality of the products or services by the customers on an online platform [51]. When high-quality information is delivered to consumers through social media platforms, they are more likely to perceive it as valuable and useable, encouraging them to make a more informed purchase decision. The excellent review comments and conversational postings on a particular subject assist consumers in locating useful information and give some critical suggestions on the subject [52]. Furthermore, once internet users come across improved quality information, they consider the content convenient and valuable [53]. By adopting the UTAUT model, researchers [54] found that IQ positively affects the usefulness and ease of use of the given information in the food delivery service application. Henceforth, based on the discussion, the study assumes that the IQ of eWOM information will determine both perceived usefulness and ease of use of eWOM information from the social media purchasing perspective. Hence, the following assumptions are proposed:

*H1*: *eWOM Information quality (IQ) will affect the perceived usefulness of eWOM information (USE).*

*H2*: *eWOM Information quality (IQ) will affect the perceived ease of use of eWOM information (EOU).*

**3.2.2. The relationship of eWOM information credibility with usefulness and ease of use of eWOM information.** In essence, credibility may be defined as the capacity to be believed from the users' perspective [55]. Information credibility can be explained as the consumers' perception and discernment of the genuineness and trustworthiness of the given information on an electronic platform such as social media [7]. Generally, information credibility indicates the extent to which a person considers the given information believable and

convincing provided by the respective electronic source [56]. The eWOM information credibility measures to what extent an online user considers the review comments or recommendations from other online users as genuine and factual [57]. Consumers usually tend to prioritize information that they feel is more believable and credible when it is available on an online platform [58], and internet users are more expected to perceive authentic information as more useful and credible [59]. The lack of credible information might result in negative customer perceptions [60]. Other authors [61] argued that the acceptance of review comments by individuals also relies on their value judgment of the credibility of the provided information in a message. Other scholars [62] advocated that eWOM has appeared as a significant medium of customer information in online purchasing. On social media platforms, online users share their viewpoints, opinions, judgments, and their negative or positive perspectives toward a particular product. This type of opinion-based eWOM information enhances the online users' perceived usefulness as they believe that the information is unbiased and credible [63]. As a result, consumers' perceptions about perceived ease of use of new information systems are influenced by reliable sources of information [64]. Therefore, the study proposes the following hypotheses:

*H3*: *Information credibility (IC) will affect the perceived usefulness of eWOM information (USE).*

*H4*: *Information credibility (IC) will affect the perceived ease of use of eWOM information (EOU).*

**3.2.3. The relationship of perceived ease of use of eWOM information (EOU) with the adoption of eWOM information (INAD) and purchase intention (PI) on social media.** The scholar [36] interpreted the idea of "perceived ease of use" as "the degree to which a person feels that utilizing a given system would be devoid of effort" (p. 320). From a consumer point of view, perceived ease of use assesses how effortless it is to utilize a specific technology-based system [65]. In the current paper, perceived ease of use of the information relates to online consumers' view that the online information may be readily and seamlessly comprehended and utilized to increase the consumer's performance in the online purchase decision-making process. Many researchers have indicated that perceived ease of use significantly impacts behavioral adoption and intention. For example, in his research study, the author [66] demonstrated that perceived ease of use positively impacted students' attitudes toward adopting mobile learning apps. The study of [67] empirically analyzed Omani students' acceptance of e-learning systems. They discovered that simplicity of use significantly impacts students' readiness to embrace e-learning systems. According to the study findings conducted by other scholars [68], ease of use is a determinant of online intention to purchase wine among online-based customers. Thus, based on the existing literature, it is believed that if eWOM information becomes easier and uncomplicated for online users on social media networks, online customers will be more likely to adopt eWOM information and will have an intention towards purchase the products or services on social media websites. Henceforth, the current paper postulates the following hypotheses:

*H5*: *Perceived ease of use of eWOM information (EOU) will affect eWOM information adoption (INAD).*

*H6*: *Perceived ease of use of eWOM information (EOU) will affect purchase intention(PI) on social media.*

**3.2.4. The relationship of perceived usefulness of information (USE) with adoption of eWOM information (INAD) and purchase intention (PI) on social media.** As defined by the author [36], "perceived usefulness" is "the degree to which a person feels that utilizing a certain system would improve his or her job performance" (p. 320). According to scholars [69], Perceived Usefulness has been defined as the user's belief that using online technology will lead to achieving their performance. According to scholars [70], the perceived usefulness of the information indicates the users' perception that new information is likely to upgrade their performance. In the case of online purchases, information adoption is driven by the degree to which the provided information is valuable on the specific online platform [71]. If online consumers consider the information, they find on social media networks to be relevant and meaningful, they are more likely to have formed a solid inclination to accept that knowledge [10]. Furthermore, prior research identified perceived usefulness as a significant predictor of online purchase intention [72–74]. Therefore, the study proposes the following hypotheses:

H7: *Perceived usefulness of eWOM information (USE) will affect eWOM information adoption (INAD).*

H8: *Perceived usefulness of eWOM information (USE) will affect purchase intention (PI) on social media.*

**3.2.5. The relationship of adoption of eWOM information (INAD) with purchase intention (PI) in social media.** The Information adoption can be referred to as a process through which individuals are purposefully involved in making use of the given information [70]. Information adoption has been regarded as one of the critical factors in impacting consumers' intention towards making a purchasing decision [16]. The adoption of eWOM information is based on the information adoption procedure that indicates the degree of the online users' utilization process of given information [75]. It is highly likely that individuals who adopt eWOM information tend to leverage it when making an actual buying decision [75]. Online users, who are likely to utilize and adopt eWOM information, tend to have buying intentions [11]. Consumers will develop a firm purchase intention if they get involved in eWOM information [76]. Several studies reported a positive connection between eWOM and purchase intention in the online platform [77–79]. Therefore, based on the discussion on the connection between eWOM information and purchase intention, the following hypothesis is postulated:

H9: *The adoption of eWOM information (INAD) will affect consumers' purchase intention (PI) on Social Media.*

Based on the above hypotheses, the following figure is proposed as the conceptual model of the study (please refer to Fig 1):

# 4. Materials and methods

From the research philosophy perspective, the positivism paradigm has been considered by the researcher to focus on facts with quantitative analysis in order to operationalize concepts so that they can be measured scientifically and systematically [80]. Therefore, it is essential to incorporate deductive methods emphasizing truly quantitative analysis and then test by empirical observation [81] to achieve predetermined well-defined research objectives.

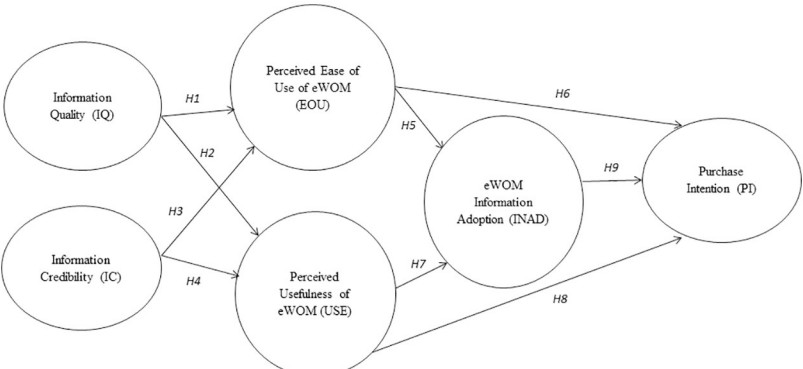

**Fig 1. Conceptual model of the study.**

## 4.1. Study population & sample

For the current investigation, university students studying in various public and private universities inside Chattogram city of Bangladesh have been considered the sampling population in this study. University students represent a diverse group of different qualities and reflect the broader community and society [82]. The convenience sampling method has been considered for this study as it allows the researcher to attain specific data and patterns without the inconvenience of a randomized sample. Additionally, this sampling method assists in capturing the presence of a certain quality of behavior within the given sample, along with identifying associations between various phenomena [83]. An online survey was conducted around May and June 2021 to obtain the required data. After eliminating 118 incomplete responses from a total of 550 questionnaires, 432 sample respondents have been chosen, resulting in an effective sample rate of about 78.55 percent. Participants were initially approached through a direct email to all university students notifying them that they had been chosen randomly to participate in this survey. Later they received an email including a link to Google form, where a brief explanation of the study was given first, and then asked to complete a questionnaire survey with demographic questions. The email also contained the researcher's contact information that students can use to reach the researcher. The online survey was highly compatible with widely used web browsers, and the surveys were easily understandable using a large font.

## 4.2. Measures

A survey instrument was developed based on earlier research which included relevant information related to social media usage and verified in two stages. Initially, the questionnaire set was delivered to researchers and professionals in the business and IT industries who have expert knowledge on such topics to ensure its simplicity and relativeness with the field. The experts' recommendations and opinions have been utilized to improve the questionnaire. Later, pilot testing was performed on a small sample of 35 respondents having university degrees and using social media-related applications, who were asked for their thoughts to minimize the complexity and ambiguity of the questionnaire. Cronbach's Alpha test was used to calculate reliability. All the variables are evaluated without any change as they all surpassed the threshold value of the Cronbach alpha reliability (0.70). In the final analysis, the data from the pilot research were not used.

   In this study, the questionnaires consisted of two distinct parts: part A is about the participants' demographic information. Part B comprises the item-wise Likert questions on

dependent and independent variables derived from various previous studies. For measuring constructs of eWOM information quality, three items were adopted from [84]; four items are used for measuring eWOM information credibility, which is adopted from [85]; three items are considered for measuring eWOM information usefulness, which is adopted from [34]; two items are considered for measuring eWOM information ease of use, which is adopted from [36]. Four items have been adopted for the measurement of eWOM information adoption [86]. Finally, four items were adopted from [87] to measure the purchase intention on social media websites. Participants were asked to rate the items prepared based on the five-point rating Likert scale (starting with strongly disagree to strongly agree).

### 4.3. Data analysis

Structural equation modeling is a well-developed scientific approach that truly emphasizes statistical tools to develop cause and effect relationships between independent and dependent constructs used for the measurement of study through developing predetermined critical hypotheses logically and analytically [88, 89]. This study has employed SPSS 23 for descriptive analysis, SMARTPLS 3.3 for confirmatory factor analysis, and a structural modeling.

## 5. Results

### 5.1. Respondents' profile

The demographic profile of respondents has been taken into consideration based on gender, age, level of education, and social media user experience (Table 1). Overall, the survey results showed that 61.81% of the participants were males, 42.13% were between the ages of 25 and 30, 56% were studying at the postgraduate level, and 63.19% had 1–3 years of usage experience of social media.

Confirmatory Factor analysis has been employed to evaluate the validity and reliability of overall and individual measurement constructs. In CFA, predetermined hypotheses are estimated to lead to a certain outcome based on an underlying causal explanation. In order to evaluate the measurement model, outer loading constructs reliability and construct validity were examined. To determine the correlation between the latent measures and the reflective indicators, the values of the outer loadings were evaluated where indicators with outer loading of 0.6 or above were considered [90]. Cronbach's alpha values were used to determine the

**Table 1. Demographic information.**

| Variables | Items | Frequency | Percent |
|---|---|---|---|
| Age | Below 20 years | 69 | 15.97% |
| | 20–25 years | 144 | 33.33% |
| | 25–30 years | 182 | 42.13% |
| | Above 30 years | 37 | 8.56% |
| Gender | Female | 165 | 38.19% |
| | Male | 267 | 61.81% |
| Education Level | Undergraduate | 187 | 43.29% |
| | Postgraduate | 245 | 56.71% |
| Social media using experience | Less than 1 year | 132 | 30.56% |
| | 1–3 years | 273 | 63.19% |
| | >3 years | 27 | 6.25% |

**n = 432

**Table 2. Construct validity results.**

| Constructs | Items | Loadings | Cronbach's Alpha | CR | AVE |
| --- | --- | --- | --- | --- | --- |
| eWOM Information quality (IQ) | IQ1 | 0.82 | 0.774 | 0.869 | 0.688 |
| | IQ3 | 0.858 | | | |
| | IQ4 | 0.809 | | | |
| eWOM Information credibility (IC) | IC1 | 0.774 | 0.77 | 0.853 | 0.592 |
| | IC2 | 0.757 | | | |
| | IC3 | 0.823 | | | |
| | IC4 | 0.72 | | | |
| eWOM Ease of use (EOU) | EOU1 | 0.844 | 0.675 | 0.859 | 0.753 |
| | EOU2 | 0.891 | | | |
| eWOM Usefulness (USE) | USE1 | 0.759 | 0.615 | 0.796 | 0.565 |
| | USE2 | 0.776 | | | |
| | USE3 | 0.72 | | | |
| eWOM Information Adoption (INAD) | INAD1 | 0.747 | 0.78 | 0.858 | 0.603 |
| | INAD2 | 0.807 | | | |
| | INAD3 | 0.827 | | | |
| | INAD4 | 0.72 | | | |
| Purchase Intention (PI) | PI1 | 0.7 | 0.793 | 0.866 | 0.619 |
| | PI2 | 0.831 | | | |
| | PI3 | 0.823 | | | |
| | PI4 | 0.787 | | | |

construct's reliability, where all values demonstrated satisfactory results, exceeding the 0.60 criterion [90]. The current study has employed convergent and discriminant validity to verify construct validity. While evaluating convergent validity, values of AVEs were between 0.565 and 0.753, and the CR values are also found within 0.796 to 0.869, which correlates to the threshold value explained by other scholars [90] (see Table 2).

## 5.2. Measurement model analysis

After concluding the convergent validity test, the discriminant validity test was performed. As shown in the literature, scholars [91] stated that for each indicator, the square root of AVE must be higher than the associated values of that specific indicator with other indicators (Table 3). Despite this, many argue that the Fornell–Larcker criteria are not justifiable since they cannot correctly evaluate the presence of discriminant validity in extensive research [92]. To address the shortcomings of the other [91] method, other authors [92] used the HTMT as a more thorough and more confined approach for researchers employing PLS-SEM to evaluate discriminant validity, specifying that the HTMT values must be less than 0.85 or 0.90 [92]. The HTMT test result is presented in Table 3, and the results met the HTMT 0.85 and HTMT 0.90 criteria, indicating that the measurement model was discriminately validated.

## 5.3. Structural model analysis

An assessment of the structural model was undertaken to determine the significance of the paths and the predictive power of the model through the PLS algorithm, and then by considering a bootstrapping using 5,000 samples process [90] (Fig 2). According to the findings (Table 4), eWOM information quality significantly influences both eWOM perceived ease of use and eWOM perceived usefulness ($\beta$ = 0.219, t-values = 4.882, significance p < 0.000 and $\beta$

**Table 3. Discriminant validity.**

Fornell and Larcker's criterion

|  | IQ | IC | EOU | USE | INAD | PI |
|---|---|---|---|---|---|---|
| IQ | 0.829 |  |  |  |  |  |
| IC | 0.371 | 0.769 |  |  |  |  |
| EOU | 0.395 | 0.557 | 0.868 |  |  |  |
| USE | 0.33 | 0.416 | 0.405 | 0.752 |  |  |
| INAD | 0.52 | 0.451 | 0.552 | 0.489 | 0.777 |  |
| PI | 0.334 | 0.375 | 0.457 | 0.517 | 0.492 | 0.787 |

Heterotrait-Monotrait Ratio (HTMT)

|  | IQ | IC | EOU | USE | INAD | PI |
|---|---|---|---|---|---|---|
| IQ |  |  |  |  |  |  |
| IC | 0.476 |  |  |  |  |  |
| EOU | 0.535 | 0.765 |  |  |  |  |
| USE | 0.482 | 0.595 | 0.624 |  |  |  |
| INAD | 0.667 | 0.575 | 0.752 | 0.701 |  |  |
| PI | 0.42 | 0.473 | 0.612 | 0.734 | 0.614 |  |

= 0.204, t-values = 3.996, significance p < 0.000), therefore confirming H1 and H2. Again, perceived ease of use and perceived usefulness of eWOM information were influenced by eWOM information credibility ($\beta$ = 0.476, t-values = 6.691, significance p < 0.000 and $\beta$ = 0.341, t-values = 5.396, significance p < 0.000), thereby validating H3 and H4. Ease of use of eWOM has shown significant influence on both eWOM information adoption and purchase intention ($\beta$ = 0.424, t-values = 8.514, significance p < 0.000 and $\beta$ = 0.203, t-values = 3.703, significance p < 0.000), and confirmed the H5 and H6.

Again, eWOM perceived usefulness had positive and significant influence on both eWOM information adoption and purchase intention ($\beta$ = 0.317, t-values = 5.913, significance p < 0.000 and $\beta$ = 0.327, t-values = 6.196, significance p < 0.000), and confirmed the H7 and H8. Finally, H9 is accepted as eWOM information adoption and purchase intention are found significant ($\beta$ = 0.221, t-values = 3.227, significance p < 0.05).

The determination coefficient ($R^2$ value) is the most broadly utilized measure to analyze the structural equation model. The $R^2$ value measures the goodness of the structural model. The

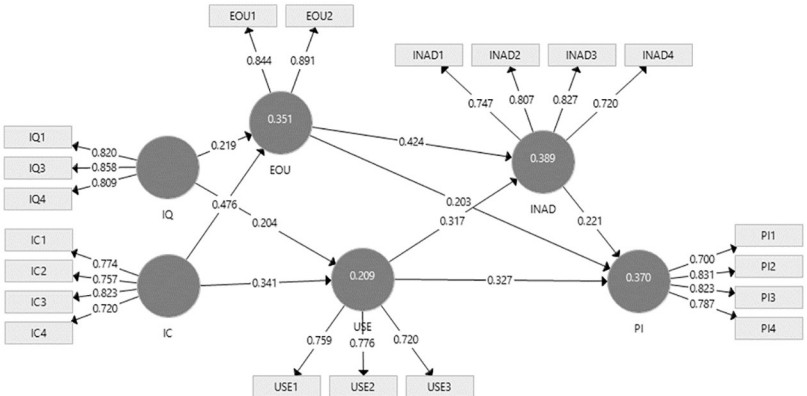

**Fig 2. Path coefficients of the model.**

**Table 4. Path coefficients.**

| Hypothesis | Path | Beta | SE | T-value | P Values | Decision |
|---|---|---|---|---|---|---|
| H1 | IQ -> EOU | 0.219 | 0.045 | 4.882 | 0.000 | Accepted |
| H2 | IQ -> USE | 0.204 | 0.051 | 3.996 | 0.000 | Accepted |
| H3 | IC -> EOU | 0.476 | 0.057 | 8.388 | 0.000 | Accepted |
| H4 | IC -> USE | 0.341 | 0.063 | 5.396 | 0.000 | Accepted |
| H5 | EOU -> INAD | 0.424 | 0.050 | 8.514 | 0.000 | Accepted |
| H6 | EOU -> PI | 0.203 | 0.055 | 3.703 | 0.000 | Accepted |
| H7 | USE -> INAD | 0.317 | 0.054 | 5.913 | 0.000 | Accepted |
| H8 | USE -> PI | 0.327 | 0.053 | 6.196 | 0.000 | Accepted |
| H9 | INAD -> PI | 0.221 | 0.068 | 3.227 | 0.001 | Accepted |

$R^2$ value of 0.37 means that the model explains that 37.0 percent of the variance in young consumers' behavior intention towards social media can be explained by the predictors in this study. Furthermore, the study assessed effect sizes ($f^2$) and predictive relevance by employing the Stone- Geisser $Q^2$ value. The "$f^2$" effect size is utilized to measure the degree to which each latent exogenous variable influences the latent endogenous variables, where the satisfactory effect size values are 0.35, 0.15, and 0.02 deemed as substantial, medium, and small effect sizes, respectively [90]. Again, research authors [90] suggested that predictive relevance $Q^2$ values should be greater than 0, which means that the exogenous variables have predictive significance over the endogenous variables and the model is predictively relevant. Table 5 summarizes the acceptable values for predictive relevance $Q^2$, effect sizes ($f^2$), and coefficient of determination $R^2$.

# 6. Discussion & implications

This study aims to examine the behavioral intention to order in social media based on the eWOM review through integrating the "Information Adoption Model (IAM)" and "Technology Acceptance Model (TAM)". Based on this, the current framework incorporates eWOM information quality, eWOM information credibility, perceived ease of use of information, perceived usefulness of information, adoption of eWOM information, and purchase intention in social media. Incorporating eWOM information quality and eWOM information credibility

**Table 5. Predictive relevance Q², effect sizes (f²), and coefficient of determination R².**

| | $R^2$ | $Q^2$ | $f^2$ | Decision |
|---|---|---|---|---|
| Purchase Intention | 0.37 | 0.219 | | |
| eWOM Information Adoption | | | 0.047 | Small |
| eWOM Information Adoption | 0.389 | 0.227 | | |
| Perceived Ease of Use of Information | | | 0.246 | Medium |
| Perceived Usefulness of Information | | | 0.138 | Small |
| Perceived Ease of Use of Information | 0.351 | 0.26 | | |
| Information Quality of Information | | | 0.064 | Small |
| Information Credibility of Information | | | 0.301 | Medium |
| Perceived Usefulness of Information | 0.209 | 0.116 | | |
| Information Quality of Information | | | 0.045 | Small |
| Information Credibility of Information | | | 0.127 | Small |

as a context creates an extended framework of TAM, which has enhanced the model's predictive ability ($R^2$ = 37%).

The eWOM information quality significantly affects information usefulness and ease of use which is relevant to the previous studies [93–95]. The eWOM information quality significantly affects information usefulness and ease of use which is relevant to the previous studies [93–95]. With regard to the online-based platform, users are more willing to take advantage of information quality if they believe that the system delivers clear, comprehensible, and current information and the content is of high quality [96]. The results from the structural equation model also indicated that the eWOM information credibility significantly affects information usefulness and ease of use which is significant to earlier studies [97]. The credibility of the provided information is crucial for improving the benefit of a delivery channel when information overflow makes selection challenging for consumers, and consumers feel that if an online platform is convenient and efficient, it has the ability to manage successful transactions [93]. Information on social media is primarily commercialized, making verifying information's quality, traceability, and credibility difficult. Therefore, reviews on social media platforms greatly assist online users in determining the traceability, authenticity, and quality of online businesses, which in turn encourage them to make a decision. In addition to the reviews provided by consumers, the firm's engagement in social media communication will help improve its credibility in affecting consumer perceptions [98].

As hypothesized, perceived ease of use had a statistically significant positive effect on behavioral intention to order in social media based on an eWOM review [97, 99]. The result shows that customers with a positive mindset towards adopting new technologies and who believe that technologies are simple to use may enhance their interest in technology adoption. And if the actual content is simple and fast retrievable, young consumers become more interested in staying on the platform. Thus, they will likely continue adopting it [97]. From the outcomes of this study, it has also been revealed that PEU positively influences users' behavioral intention, which is consistent with previous research studies [73, 100]. Young consumers will enjoy sharing eWOM on social networks if the social media content is simple and easy to use, combines consumer-focused technology with socialization patterns, and doesn't limit capabilities for users to post, share, and maintain relationships [2].

Perceived usefulness is significantly related to information adoption, which is significant to previous studies [101–103]. The perceived usefulness of harnessing social media in accomplishing tasks may be interpreted as the adoption of information by users that would help them better complete their tasks [104]. Information that is useful is considered the critical determining factor influencing individuals to adopt that information [101, 102]. Again, perceived usefulness is significantly associated with behavioral intention [105]. In particular, while communicating on the eWOM platform, its usefulness substantially influences people's perception of using technology and their intention to adopt eWOM [106]. Individuals using social networks are introduced to a large volume of eWOM information, which might lead to a higher inclination to incorporate information that seems beneficial to them [71].

Furthermore, it has been noted that information adoption is significantly and positively related to online purchase intention, which supports earlier studies [71, 107]. Information adoption is commonly acknowledged as something that will enable or hinder decisions [108]. The usefulness of information is crucial in purchasing decisions since it aids consumers in determining and understanding products, services, and associated aspects of their purchasing decisions [102]. When individuals adopt information about the rival brand and the attributes and benefits of a product or service, this enhances their buying decisions since it facilitates them in evaluating their options [107]. The finding revealed that information adoption partly mediates the suggested relationship [107].

In comparison to direct effects, the mediation analysis revealed that the impact of perceived ease of use and perceived usefulness on behavioral intention was mediated by information adoption. Although the contrast between the direct effects of perceived ease of use and perceived usefulness on behavioral intention and the indirect effects through information adoption indicates that the indirect effect was lower, however, the partial mediation and significant relationship reveal that information adoption has a significant role in affecting consumers' behavioral intention. Overall, the results of the current study point to the importance of eWOM contents in social media, i.e., quality and credibility, as well as to perceived usefulness and ease of use associated with information adoption, which eventually affects behavioral intention toward ordering in social media based on eWOM review.

The current research findings offer the marketing managers a viewpoint to realize the significance of the effect of eWOM information on online purchase intention among social media users. As this research reports that online purchase intention among social media users is connected with eWOM information, hence it does indicate that marketers create brands by leveraging social media platforms. The research findings suggest that both eWOM information credibility and quality enhance the ease of use and usefulness of eWOM information that would affect adoption and purchase intention. So, marketers need to place reliable and quality information about their products or services on social media websites so that it might lead to developing a better impression in online consumers' mindsets. To enhance the online consumers' likelihood to purchase, marketers should focus on uploading useful and understandable content to promote their product/services on social media websites to enhance online consumers' propensity to buy the product. Social media users usually seek more information before making a purchase decision online; thus, business firms may think of developing fan pages to share the latest update about product/service and presenting genuine and complete information to online consumers on social media websites. Furthermore, business managers and marketers should close track and monitor the consumer-generated eWOM on social media to keep themselves updated about consumers' current viewpoints. To do this, business firms can hire online customer managers, who will assist online consumers by providing helpful, authentic, trustworthy, quality, and intelligible information as per consumer's requirements so that consumers would provide positive eWOM about the company or product, which will affect other consumers' purchasing behaviors on social media websites.

## 7. Conclusions & directions for future research

The current paper has set out the objective to explore the influence of various aspects of eWOM information purchase intention on social media websites by considering relevant components from IAM and TAM models. The study's research model has been validated through PLS-SEM, where social media using 432 university students participated as respondents. The research findings indicate that quality, credibility, usefulness, and ease of use of eWOM information have been critical in determining online consumers' intention to adopt eWOM and form purchase behavior on social media. From the managerial perspective, it has been pointed out that marketing and business managers should utilize social media websites to gauge consumer behavior by focusing on characteristics of eWOM information on social media to obtain better consumer insights. Nonetheless, the present research has few limitations. Hence, the research findings of this paper should be considered by taking into account the following research limitations. Firstly, the study developed a research model and measured the model using the PLS-SEM method, which is a quantitative approach. Future studies may consider a qualitative approach to find some new characteristics of eWOM information which may influence online consumer behavior. Secondly, the sample was drawn from social media using

university students in Bangladesh, which might not represent the entire population. Thus, considering samples from different age groups in future studies would be more helpful in producing more reflective and generalizable findings. Also, age can be regarded as a moderating variable to capture the differential age effect on the relationship between eWOM information and online consumer behavior. Finally, the study has drawn samples from Bangladeshi university students; hence future studies may adopt samples from developing and developed countries to understand the influence of cultural differences on the relationship between determinants of eWOM information and social media consumer behavior. Also, to generate a greater degree of generalization of the study findings, the current study has not solely focused on a particular product category which might also be considered a limitation of the study. Future studies might consider the specific product category to reveal the impact of eWOM-related characteristics on consumer buying behavior.

## Supporting information

**S1 Dataset.**
(CSV)

## Author Contributions

**Conceptualization:** H. M. Kamrul Hassan, Ahmed Al Asheq.

**Data curation:** Md. Atikur Rahaman, H. M. Kamrul Hassan.

**Formal analysis:** Md. Atikur Rahaman, H. M. Kamrul Hassan, Ahmed Al Asheq.

**Funding acquisition:** Md. Atikur Rahaman.

**Investigation:** H. M. Kamrul Hassan.

**Methodology:** H. M. Kamrul Hassan, Ahmed Al Asheq.

**Project administration:** Md. Atikur Rahaman, H. M. Kamrul Hassan.

**Resources:** Md. Atikur Rahaman.

**Software:** Md. Atikur Rahaman, H. M. Kamrul Hassan.

**Supervision:** H. M. Kamrul Hassan.

**Validation:** Md. Atikur Rahaman, H. M. Kamrul Hassan, K. M. Anwarul Islam.

**Visualization:** Md. Atikur Rahaman, H. M. Kamrul Hassan, K. M. Anwarul Islam.

**Writing – original draft:** H. M. Kamrul Hassan, Ahmed Al Asheq.

**Writing – review & editing:** Md. Atikur Rahaman, Ahmed Al Asheq, K. M. Anwarul Islam.

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
