## [Decision Letter · Decision Letter 0]

24 Jan 2022

PONE-D-21-35683The Interplay between eWOM Information and Purchase In-tention on Social Media: Through the Lens of IAM and TAM TheoryPLOS ONE

Dear Dr. ASHEQ,

Thank you for submitting your manuscript to PLOS ONE. After careful consideration, we feel that it has merit but does not fully meet PLOS ONE’s publication criteria as it currently stands. Therefore, we invite you to submit a revised version of the manuscript that addresses the points raised during the review process. Please follow suggestions from Reviewers and please submit your revised manuscript by Mar 09 2022 11:59PM. If you will need more time than this to complete your revisions, please reply to this message or contact the journal office at plosone@plos.org. Please include the following items when submitting your revised manuscript:A rebuttal letter that responds to each point raised by the academic editor and reviewer(s). You should upload this letter as a separate file labeled 'Response to Reviewers'.A marked-up copy of your manuscript that highlights changes made to the original version. You should upload this as a separate file labeled 'Revised Manuscript with Track Changes'.An unmarked version of your revised paper without tracked changes. You should upload this as a separate file labeled 'Manuscript'.

We look forward to receiving your revised manuscript.

Kind regards,

Jarosław Jankowski

Academic Editor

PLOS ONE

Journal Requirements:

5. Please ensure that you refer to Figure 1 in your text as, if accepted, production will need this reference to link the reader to the figure.

Reviewers' comments:

Reviewer's Responses to Questions

**Comments to the Author**

1. Is the manuscript technically sound, and do the data support the conclusions?

Reviewer #1: Yes

Reviewer #2: Yes

2. Has the statistical analysis been performed appropriately and rigorously? 

Reviewer #1: Yes

Reviewer #2: Yes

3. Have the authors made all data underlying the findings in their manuscript fully available?

Reviewer #1: No

Reviewer #2: No

4. Is the manuscript presented in an intelligible fashion and written in standard English?

Reviewer #1: Yes

Reviewer #2: Yes

5. Review Comments to the Author

Reviewer #1: This is a well-designed study with a solid theoretical foundation and reliable findings.

The problem statement of this study is not convincing. The authors state that "aspects of eWOM information and online purchase intention on social media sites are not well explored in the literature”, but fail to explain in detail the novelty of this study. According to the search, various studies have been conducted to examine the characteristics of eWOM, or customer reviews, and also their impact on customer behavior. Furthermore, the cited literature supporting the necessarity is outdated (Cheung and Thadani, 2012). The authors should clearly explain how this study differs from previous studies.

The rest part of this study is well written, systematically presenting the theoretical background of the research and the relevant concepts. Also the research process is very scientific. However, I personally do not feel that the contribution of the research results is very significant. Firstly, the study is not specific to a particular industry or related product, the adoption criteria for eWom may vary greatly with the product or industry; secondly, the sample of the study is only limited to students.

Overall, this is a well-designed study with excellent writing. The main limitation is that the contribution of this research is not very significant.

Reviewer #2: Overall the paper was written in good English but with some mistakes.

There are errors in citation, for e.g.

D.-H. Park, H. Lee and some others.

advised by (Hair Jr et al., 2021)

to order?

Overall is a OK paper but not exciting.

6. PLOS authors have the option to publish the peer review history of their article (what does this mean?). If published, this will include your full peer review and any attached files.

Reviewer #1: **Yes: **Kai Ding

Reviewer #2: **Yes: **Lai Soon, Wong

---

## [Author Response · Author response to Decision Letter 0]

11 Mar 2022

Reviewer #1: This is a well-designed study with a solid theoretical foundation and reliable findings.

The problem statement of this study is not convincing. The authors state that "aspects of eWOM information and online purchase intention on social media sites are not well explored in the literature”, but fail to explain in detail the novelty of this study. According to the search, various studies have been conducted to examine the characteristics of eWOM, or customer reviews, and also their impact on customer behavior. Furthermore, the cited literature supporting the necessarity is outdated (Cheung and Thadani, 2012). The authors should clearly explain how this study differs from previous studies.

The rest part of this study is well written, systematically presenting the theoretical background of the research and the relevant concepts. Also the research process is very scientific. However, I personally do not feel that the contribution of the research results is very significant. Firstly, the study is not specific to a particular industry or related product, the adoption criteria for eWom may vary greatly with the product or industry; secondly, the sample of the study is only limited to students.

Overall, this is a well-designed study with excellent writing. The main limitation is that the contribution of this research is not very significant.

Our response: The study's problem statement, contribution, and research gap have been strengthened with recent citations (kindly refer to page 3 to 4, yellow marked). In the marked yellow portion of page 3, 09 (nine) latest citations are provided to enhance the quality of the paper. The paper does not focus on any specific product category as this can be perceived as one of the limitations of this study (please refer to page 22, yellow marked). Regarding the sampling, we’ve also considered this as our limitation as we drew samples only from the students (please refer to page 22, blue colored). The study’s contribution is exhibited in the page 3 & 4 (yellow marked).

Reviewer #2: Overall the paper was written in good English but with some mistakes. There are errors in citation, for e.g. D.-H. Park, H. Lee and some others. advised by (Hair Jr et al., 2021) to order?

Overall is a OK paper but not exciting.

Our response: We have corrected all in-text citations throughout the manuscript (please refer to yellow marked in-text citations in page 6, 7, 12, 13. The study's problem statement, contribution, and research gap have been strengthened with recent citations (kindly refer to page 3 to 4, yellow marked). In the marked yellow portion of page 3, 09 (nine) latest citations are provided to enhance the quality of the paper.

Reviewers' comments: Have the authors made all data underlying the findings in their manuscript fully available? 

Our response: Relevant data sets (1. Raw data file_csv file, 2. Measurement model_xlsx file, 3. Structural model__xlsx file) have been provided with the revised manuscript.

---

## [Decision Letter · Decision Letter 1]

17 Jun 2022

PONE-D-21-35683R1The Interplay between eWOM Information and Purchase In-tention on Social Media: Through the Lens of IAM and TAM TheoryPLOS ONE

Dear Dr. ASHEQ,

Thank you for submitting your manuscript to PLOS ONE. After careful consideration, I feel that it has merit but does not fully meet PLOS ONE’s publication criteria as it currently stands. 

In view of the referees’ feedback and my own reading of your paper, we invite you to address all issues noted below, most of which are relatively minor in nature, but nonetheless essential. In particular, the paper needs a copy-editing process to avoid some confusion sentences and minor errors.

After that I think al major issues have been addressed and the paper would be ready for publication.

We look forward to receiving your revised manuscript.

Kind regards,

J E. Trinidad Segovia

Section Editor

PLOS ONE

Journal Requirements:

Reviewers' comments:

Reviewer's Responses to Questions

**Comments to the Author**

1. If the authors have adequately addressed your comments raised in a previous round of review and you feel that this manuscript is now acceptable for publication, you may indicate that here to bypass the “Comments to the Author” section, enter your conflict of interest statement in the “Confidential to Editor” section, and submit your "Accept" recommendation.

Reviewer #1: All comments have been addressed

Reviewer #2: (No Response)

2. Is the manuscript technically sound, and do the data support the conclusions?

Reviewer #1: Yes

Reviewer #2: Partly

3. Has the statistical analysis been performed appropriately and rigorously? 

Reviewer #1: I Don't Know

Reviewer #2: Yes

4. Have the authors made all data underlying the findings in their manuscript fully available?

Reviewer #1: Yes

Reviewer #2: No

5. Is the manuscript presented in an intelligible fashion and written in standard English?

Reviewer #1: Yes

Reviewer #2: No

6. Review Comments to the Author

Reviewer #1: (No Response)

Reviewer #2: There are some grammatical problems in this paper, it should be edited first before submit for our review. For e.g. :

The dearth of credible information might lead to form negative impressions among customers.

Also, credible source of information also affects users’ perceived ease of use in regard to adopt a new information system

Quite an ordinary paper, didn't see any contribution from this work.

And many others.

There are also some redundance sentences. All these problems shown that the writer did not do the work rigorously.

I have no problem if Plos One want to accept this paper but personally I think it needs to be revise thoroughly before it is published.

My opinion is either reject or revise thoroughly before submit for review.

7. PLOS authors have the option to publish the peer review history of their article (what does this mean?). If published, this will include your full peer review and any attached files.

Reviewer #1: No

Reviewer #2: No

---

## [Author Response · Author response to Decision Letter 1]

7 Jul 2022

We have addressed all the review comments in the response letter.

Reviewer 1:

No. Comments of Reviewer-01 Our Responses

1. If the authors have adequately addressed your comments raised in a previous round of review and you feel that this manuscript is now acceptable for publication, you may indicate that here to bypass the “Comments to the Author” section, enter your conflict of interest statement in the “Confidential to Editor” section, and submit your "Accept" recommendation. 

Reviewer-1’s comment: All comments have been addressed. 

 Thank you so much for your positive compliment.

2. Is the manuscript technically sound, and do the data support the conclusions? 

Reviewer-1’s comment: Yes

 Many thanks for your positive feedback.

3. Has the statistical analysis been performed appropriately and rigorously?

Reviewer-1’s comment: I Don't Know We have utilized deductive research approach as this method is suitable for empirical study (please refer to first para of “section 4. Materials and Method”, page 10). Our sample size is 432 which is statistically deemed as acceptable sample size for quantitatve study (Burns et al., 1995). We have described our development of survey instrument is reported in the 2nd paragraph in page 11. We have also reported convergent and discriminant validity to verify construct reliability and validity (page 13-15). 

Burns, A. C., Burns, K. R., & Bush, R. F. (1995, March). The SPSS Student Assistant: The Integration of a Statistical Analysis Program into a Marketing Research Textbook. In Developments in Business Simulation and Experiential Learning: Proceedings of the Annual ABSEL conference (Vol. 22). 

4. Have the authors made all data underlying the findings in their manuscript fully available?

Reviewer-1’s comment: Yes Thank you so much for your positive compliment.

5. Is the manuscript presented in an intelligible fashion and written in standard English?

Reviewer-1’s comment: Yes

 Thank you so much for your positive compliment.

6. Please use the space provided to explain your answers to the questions above. You may also include additional comments for the author, including concerns about dual publication, research ethics, or publication ethics. (Please upload your review as an attachment if it exceeds 20,000 characters). 

Reviewer-1’s comment: (No Response) Thank you. 

Reviewer 2:

No. Comments of Reviewer-02 Our Responses

1. If the authors have adequately addressed your comments raised in a previous round of review and you feel that this manuscript is now acceptable for publication, you may indicate that here to bypass the “Comments to the Author” section, enter your conflict of interest statement in the “Confidential to Editor” section, and submit your "Accept" recommendation. 

Reviewer-2’s comment: (No Response)

 Thank you.

2. Is the manuscript technically sound, and do the data support the conclusions? 

Reviewer-2’s comment: Partly

 Thank you so much for your feedback. We have utilized deductive research approach as this method is suitable for empirical study (please refer to first para of “section 4. Materials and Method”, page 10). Our sample size is 432 which is statistically deemed as acceptable sample size for quantitatve study (Burns et al., 1995). We have described our development of survey instrument is reported in the 2nd paragraph in page 11. We have also reported convergent and discriminant validity to verify construct reliability and validity (page 13-15). 

Burns, A. C., Burns, K. R., & Bush, R. F. (1995, March). The SPSS Student Assistant: The Integration of a Statistical Analysis Program into a Marketing Research Textbook. In Developments in Business Simulation and Experiential Learning: Proceedings of the Annual ABSEL conference (Vol. 22). 

3. Has the statistical analysis been performed appropriately and rigorously?

Reviewer-2’s comment: Yes Thank you so much for your positive compliment.

4. Have the authors made all data underlying the findings in their manuscript fully available?

Reviewer-2’s comment: No Thank you for your comment. Relevant data sets (1. Raw data file_csv file, 2. Measurement model_xlsx file, 3. Structural model__xlsx file) have been uploaded in a zip file in the submission platform with the revised manuscript.

5. Is the manuscript presented in an intelligible fashion and written in standard English?

Reviewer-2’s comment: No

 This time, we have taken professional proof read service from a native English scholar to improve the English quality of our revised manuscript (please refer to the yellow mark across the manuscript that reflects the English proof read. 

Reviewer-2’s comment: There are some grammatical problems in this paper, it should be edited first before submit for our review. For e.g. : The dearth of credible information might lead to form negative impressions among customers. Also, credible source of information also affects users’ perceived ease of use in regard to adopt a new information system.

Quite an ordinary paper, didn't see any contribution from this work. And many others. There are also some redundance sentences. All these problems shown that the writer did not do the work rigorously. 

I have no problem if Plos One want to accept this paper but personally I think it needs to be revise thoroughly before it is published. My opinion is either reject or revise thoroughly before submit for review. Thank you for your comment. We have further taken the English proof-reading service from one academic scholar who is a born English native speaker to improve the English language of our manuscript. 

We have carefully edited the entire manuscript. The indicated sentence is now written as “The lack of credible information might result in negative customer perception” on page 7 (yellow mark).

Also, the second indicated sentence is now written as “consumers’ perceptions about perceived ease of use of new information systems are influenced by reliable sources of information” on pages 7-8 (yellow mark).

 The novelty of our research is reported in the second paragraph on page 3. 

NOTE: For This study was supported by the School of Management, Jiujiang University, China. There is no grant number issued by the department. In regard to data availability statement, we’ve provided all necessary data files with the manuscript (please refer to 1. Raw data file_csv file, 2. Measurement model_xlsx file, 3. Structural model__xlsx file).

---

## [Decision Letter · Decision Letter 2]

18 Jul 2022

PONE-D-21-35683R2The Interplay between eWOM Information and Purchase Intention on Social Media: Through the Lens of IAM and TAM TheoryPLOS ONE

Dear Dr. ASHEQ,

Thank you for submitting your manuscript to PLOS ONE. After careful consideration, I feel that it has merit but does not fully meet PLOS ONE’s publication criteria as it currently stands. There are not concerns  regarding to the research showed in the manuscript but some minor details need to be corrected.

We look forward to receiving your revised manuscript.

Kind regards,

J E. Trinidad Segovia

Section Editor

PLOS ONE

Journal Requirements:

Reviewers' comments:

Reviewer's Responses to Questions

**Comments to the Author**

1. If the authors have adequately addressed your comments raised in a previous round of review and you feel that this manuscript is now acceptable for publication, you may indicate that here to bypass the “Comments to the Author” section, enter your conflict of interest statement in the “Confidential to Editor” section, and submit your "Accept" recommendation.

Reviewer #2: (No Response)

2. Is the manuscript technically sound, and do the data support the conclusions?

Reviewer #2: Yes

3. Has the statistical analysis been performed appropriately and rigorously? 

Reviewer #2: Yes

4. Have the authors made all data underlying the findings in their manuscript fully available?

Reviewer #2: Yes

5. Is the manuscript presented in an intelligible fashion and written in standard English?

Reviewer #2: Yes

6. Review Comments to the Author

Reviewer #2: There are still some mistakes as pointed out in the attached text attached. Some suggestions been provided too. Is up to you to decide whether you want to correct or not. I can't say I am 100% correct. Is up to you and up to Plos One to accept your paper. I don't have any other concern of recommending the acceptance of your paper. Whatever written is a reflection of your work. You may contact me if you want at lswong@utar.edu.my

7. PLOS authors have the option to publish the peer review history of their article (what does this mean?). If published, this will include your full peer review and any attached files.

Reviewer #2: **Yes: **Wong Lai Soon

---

## [Author Response · Author response to Decision Letter 2]

19 Jul 2022

Reviewer 2:

No. Comments of Reviewer-02 Our Responses

1. If the authors have adequately addressed your comments raised in a previous round of review and you feel that this manuscript is now acceptable for publication, you may indicate that here to bypass the “Comments to the Author” section, enter your conflict of interest statement in the “Confidential to Editor” section, and submit your "Accept" recommendation. 

Reviewer-2’s comment: (No Response)

 Thank you.

2. Is the manuscript technically sound, and do the data support the conclusions? 

Reviewer-2’s comment: Yes

 Thank you so much for your positive acceptance. 

3. Has the statistical analysis been performed appropriately and rigorously?

Reviewer-2’s comment: Yes

 Thank you so much for your positive compliment.

4. Have the authors made all data underlying the findings in their manuscript fully available?

Reviewer-2’s comment: Yes Thank you so much for your positive compliment..

5. Is the manuscript presented in an intelligible fashion and written in standard English?

Reviewer-2’s comment: Yes

 Thank you so much for your positive compliment.

6. Review Comments to the Author

Please use the space provided to explain your answers to the questions above. You may also include additional comments for the author, including concerns about dual publication, research ethics, or publication ethics. (Please upload your review as an attachment if it exceeds 20,000 characters)\\

Reviewer #2: There are still some mistakes as pointed out in the attached text attached. Some suggestions been provided too. Is up to you to decide whether you want to correct or not. I can't say I am 100% correct. Is up to you and up to Plos One to accept your paper. I don't have any other concern of recommending the acceptance of your paper. Whatever written is a reflection of your work. You may contact me if you want at lswong@utar.edu.my

 Thank you so much for your kind feedback in the attached manuscript. We have tried our best to correct the grammatical issues in the manuscript (kindly refer to yellow marked sections). 

NOTE: This study was supported by the School of Management, Jiujiang University and Center for Entreprenureship & Innovation Research Society (grant number: 43120227). The funders have no role in study design, data collection and analysis, decision to publish, or preparation of the manuscript. In regard to data availability statement, we’ve provided all necessary data files with the manuscript (please refer to 1. Raw data file_csv file, 2. Measurement model_xlsx file, 3. Structural model__xlsx file).

---

## [Decision Letter · Decision Letter 3]

29 Jul 2022

The Interplay between eWOM Information and Purchase Intention on Social Media: Through the Lens of IAM and TAM Theory

PONE-D-21-35683R3

Dear Dr. ASHEQ,

We’re pleased to inform you that your manuscript has been judged scientifically suitable for publication and will be formally accepted for publication once it meets all outstanding technical requirements.

Kind regards,

J E. Trinidad Segovia

Section Editor

PLOS ONE

Additional Editor Comments (optional):

Reviewers' comments:

Reviewer's Responses to Questions

**Comments to the Author**

1. If the authors have adequately addressed your comments raised in a previous round of review and you feel that this manuscript is now acceptable for publication, you may indicate that here to bypass the “Comments to the Author” section, enter your conflict of interest statement in the “Confidential to Editor” section, and submit your "Accept" recommendation.

Reviewer #2: All comments have been addressed

2. Is the manuscript technically sound, and do the data support the conclusions?

Reviewer #2: Yes

3. Has the statistical analysis been performed appropriately and rigorously? 

Reviewer #2: Yes

4. Have the authors made all data underlying the findings in their manuscript fully available?

Reviewer #2: Yes

5. Is the manuscript presented in an intelligible fashion and written in standard English?

Reviewer #2: Yes

6. Review Comments to the Author

Reviewer #2: Although not all comments were addressed but this paper is in an acceptable format. I think it can be accepted now.

7. PLOS authors have the option to publish the peer review history of their article (what does this mean?). If published, this will include your full peer review and any attached files.

Reviewer #2: **Yes: **Wong Lai Soon

---

## [Editor Report · Acceptance letter]

25 Aug 2022

PONE-D-21-35683R3 

The Interplay between eWOM Information and Purchase Intention on Social Media: Through the Lens of IAM and TAM Theory 

Dear Dr. Asheq:

I'm pleased to inform you that your manuscript has been deemed suitable for publication in PLOS ONE. Congratulations! Your manuscript is now with our production department. 

Kind regards, 

on behalf of

Dr. J E. Trinidad Segovia 

Section Editor

PLOS ONE